# Blood Flow Restriction in Oncological Patients: Advantages and Safety Considerations

**DOI:** 10.3390/healthcare11142062

**Published:** 2023-07-19

**Authors:** Maria Jesus Vinolo-Gil, Ismael García-Campanario, María-José Estebanez-Pérez, José-Manuel Pastora-Bernal, Manuel Rodríguez-Huguet, Francisco Javier Martín-Vega

**Affiliations:** 1Department of Nursing and Physiotherapy, University of Cadiz, 11003 Cadiz, Spain; manuel.rodriguez@uca.es (M.R.-H.); javier.martin@uca.es (F.J.M.-V.); 2Institute for Biomedical Research and Innovation of Cádiz, 11009 Cadiz, Spain; 3Rehabilitation Clinical Management Unit, Interlevels-Intercenters Hospital Puerta del Mar, Hospital Puerto Real, Cadiz Bay-La Janda Health District, 11006 Cadiz, Spain; 4Department of Medicine, Faculty of Medicine, University of Cadiz, Grupo PAIDI UCA CTS391, 11003 Cadiz, Spain; ismael.garcia@uca.es; 5Department of Physiotherapy, Faculty of Health Science, University of Malaga, 29071 Malaga, Spain; mariajoseestebanezperez@uma.es (M.-J.E.-P.); jmpastora@uma.es (J.-M.P.-B.)

**Keywords:** blood flow restriction, KAATSU training, blood occlusion, blood flow restriction exercise, cancer, neoplasms, cancer survivors, oncology

## Abstract

Background: Cancer, being a highly widespread disease on a global scale, has prompted researchers to explore innovative treatment approaches. In this regard, blood flow restriction has emerged as a promising procedure utilized in diverse clinical populations with favorable results including improvements in muscle strength, cardiovascular function, and postoperative recovery. The aim of this systematic review was to assess the efficacy of blood flow restriction in cancer survivors. Methods: An investigation was carried out using various databases until February 2023: PubMed, Scientific Electronic Library Online, Physiotherapy Evidence Database, Scopus, Web of Science, Cochrane Plus, SPORTDiscus, Physiotherapy and Podiatry of the Complutense University of Madrid, ScienceDirect, ProQuest, Research Library, Cumulative Index of Nursing and Allied Literature Complete Journal Storage, and the gray literature. To assess the methodological quality of the studies, the PEDro scale was utilized, and the Cochrane Collaboration tool was employed to evaluate the risk of bias. Results: Five articles found that blood flow restriction was beneficial in improving several factors, including quality of life, physical function, strength, and lean mass, and in reducing postoperative complications and the length of hospital stay. Conclusion: Blood flow restriction can be a viable and effective treatment option. It is important to note that the caution with which one should interpret these results is due to the restricted quantity of articles and significant variation, and future research should concentrate on tailoring the application to individual patients, optimizing load progression, ensuring long-term follow-up, and enhancing the methodological rigor of studies, such as implementing sample blinding.

## 1. Introduction

Cancer is a disease that is widespread globally and affects millions of individuals on an annual basis. The World Health Organization reports that cancer is among the primary causes of morbidity and mortality worldwide, with an estimated 19.3 million new cases and 10 million cancer-related deaths in 2020 [1]. It is anticipated that the incidence of cancer will increase in the coming years, highlighting the need for effective interventions that can improve patient outcomes and quality of life [2].

Cancer and its treatments can cause a range of physical and emotional symptoms, including fatigue, decreased muscle mass, and weakness. These symptoms can significantly impact the quality of life of oncological patients, affecting their capacity to perform everyday tasks and engage in physical exercise [3]. 

Although cancer and its treatments can cause severe symptoms, research indicates that exercise can significantly alleviate them [4] by helping to reduce the risk of cancer, slow the progression of the disease, assist with antineoplastic treatments, and improve patients’ physical and mental health [5,6,7]. Exercise has been shown to enhance physical ability and to decrease fatigue, improving the overall quality of life of cancer survivors [8]. Additionally, regular exercise can aid in preventing or reducing the muscle wasting associated with cancer and its treatments [9].

Furthermore, it has been indicated that exercise can prevent the recurrence of cancer, reducing the risk of cancer-related mortality and improving the overall survival rates of cancer patients [10]. According to a meta-analysis carried out by Mustian et al. [11], exercise was found to be among the most effective therapies for cancer-related fatigue compared to pharmaceutical or psychological interventions. The American College of Sports Medicine also includes exercise guidelines for cancer patients, including both aerobic and resistance training [12].

Regardless of these benefits, many oncological patients are hesitant to engage in workout due to concerns about exacerbating the symptoms or causing harm to their health. There is strong evidence that moderate-intensity aerobic training or resistance exercise during cancer treatment and post-treatment reduces anxiety, depression, and fatigue and improves health-related quality of life and physical functioning [13].

Considering the potential benefits and challenges of fitness for these patients, it is crucial to design exercise programs that are safe, effective, and tailored to their unique needs [9]. One promising rehabilitation strategy that could address these needs is blood flow restriction (BFR), which utilizes cuffs or other devices to partially decrease the blood supply to the muscles during training. This technique involves applying a pneumatic pressure cuff around the upper portion of the targeted limb, which is kept inflated during the session to restrict blood flow to that area. Research has indicated that BFR training is more efficient in preventing muscle wasting and weakness caused by immobilization and unloading compared to isometric exercise alone [14]. Additionally, such an approach has been utilized alongside aerobic or strength training to enhance muscular hypertrophy and strength in healthy individuals [15,16] and athletes [17] while reducing the risk of injury [18]. Therefore, BFR training seems to be a viable choice for addressing the early stages of rehabilitation when the muscles may not tolerate higher loads [19].

BFR has also been used to improve muscle function in various clinical populations, such as people with lung disease [20], postmenopausal females [21], in Acquired Immune Deficiency Syndrome (AIDS) [22], the geriatric population [23,24], chronic kidney disease [25], heart failure and heart disease [26], diabetes [27], neurological diseases [28,29,30], and musculoskeletal disorders [18,31]. In recent years, BFR has gained interest as a potential intervention for cancer patients, who commonly experience muscle weakness and fatigue due to cancer treatment [32]. Also, it enhances physical staying power by increasing the supply of oxygen to muscle tissue [33].

There is limited investigation on the use of BFR training specifically for cancer patients. However, some articles have examined the potential advantages of BFR in other groups with muscle weakness or wasting [23,24], which may have implications for cancer patients.

BFR exercise therapy for elderly individuals with weak muscles has resulted in improvements in the power of muscles, muscular volume, and physical performance [14]. Other studies examined the use BFR training in individuals with chronic obstructive pulmonary disease (COPD) and found improvements in muscle strength and staying power [20,34]. COPD and cancer patients often experience similar symptoms, including muscle weakness and wasting, so these findings may also have relevance for cancer patients.

Despite the limited study on the use of BFR training in individuals affected by cancer, some experts suggest that it may be a promising strategy for improving muscle strength and function in this population [35]. However, further studies are required to establish the safety and effectiveness of BFR for cancer survivors.

This review aims to investigate the effects of BFR training in cancer patients by addressing the following research questions: (1) What are the physiological and functional benefits of BFR training in oncological patients? (2) How does BFR training impact muscle strength, endurance, and hypertrophy in cancer patients? (3) Are there any potential safety concerns or adverse effects associated with BFR training in this population?

## 2. Materials and Methods

### 2.1. Search Strategy

An investigation for scientific evidence was conducted following the PRISMA re-commendations for systematic reviews [36]; the investigation methodology was recorded in the systematic review database PROSPERO (CRD42022354827). The bibliographic research was conducted until February 2023 using the scientific databases Cumulative Index of Nursing and Allied Literature Complete (CINAHL), PubMed, Physiotherapy Evidence Database (PEDro), Scopus, Scientific Electronic Library Online (SciELO), Cochrane Plus, SPORTDiscus, Physiotherapy and Podiatry of the Complutense University of Madrid (ENFISPO), Web of Science (WOS), ScienceDirect, ProQuest, Research Library, and Journal Storage (JSTOR), as well as gray literature (TESEO, OpenGrey, and Grey Literature Database). We used the search strategy shown in Table 1.

The following keywords were used in combination with the Boolean operators AND and OR: “cancer”, “neoplasm”, “malignant”, “carcinoma”, “tumor”, “tumor”, “oncology”, “blood flow restriction therapy”, “limb occlusion pressure”, “blood flow restriction”, “blood flow restriction exercise”, “BFR exercise”, “KAATSU training”, “limb occlusion”, “vascular restriction”, and “KAATSU”.

In the investigation method, all accessible documents without language restriction were encompassed without any filtration based on study type or publication date. Non-English articles were translated into English to ensure a comprehensive analysis. Various study types, such as randomized trials, observational studies, and case studies, were included to gather a comprehensive understanding of the topic. By incorporating different study types, the review aimed to explore a wide range of evidence while accounting for their respective strengths and limitations.

### 2.2. Eligibility Criteria

The documents included in the systematic review were selected based on specific inclusion and exclusion criteria. The research question followed the “PICOS” model [37]: Participants (P): Patients with a diagnosis of any type of cancer; Intervention (I): Blood flow restriction (BFR); Comparison (C): Absence of treatment, placebo, or alternative strategy; Outcome (O): Any physical or psychological factor that has the potential for enhancement or betterment or any variable related to the use of blood flow restriction; Study design (S): Randomized and non-randomized clinical trials, observational studies, or case studies.

There were no limitations based on the publication date, language of the articles, or patients’ age. However, studies that examined circulatory changes unrelated to the BFR intervention were excluded.

### 2.3. Evaluation of Methodological Rigor

Two independent reviewers (M.J.V.-G. and M.-J.E.-P.) evaluated the methodological quality of the randomized controlled trials (RCTs) included in this study using the ROB 2 (Risk of Bias 2) tool. The ROB 2 tool is widely recognized in the scientific literature for assessing the risk of bias in RCTs and provides a systematic assessment of the different domains of methodological quality, such as the randomization process, allocation concealment, blinding, data integrity, and selective outcome reporting [38].

Disagreements between the authors were initially resolved through discussion and finally by consultation with a third reviewer (F.J.M.-V.).

On the other hand, to assess the methodological quality of the observational studies included, the STROBE (Strengthening the Reporting of Observational Studies in Epidemiology) scale was employed. The STROBE scale is a widely recognized tool used to evaluate the reporting quality and methodological rigor of observational studies. It consists of a checklist of 22 items, covering key aspects such as study design, participant selection, variables measured, statistical methods, and interpretation of results. Each item was assessed for its presence or adequacy in the studies included, and a percentage score was calculated based on the number of fulfilled recommendations [39].

### 2.4. Selection and Data Extraction Process

The process of selecting studies included several stages. Initially, the main database, PubMed, was searched to obtain appropriate descriptors, which were then used to search all of the mentioned databases, as outlined in Table 1. The selected studies’ data included author, year of publication, participant demographics (overall sample size and group-wise distribution), intervention details (exercise modality, sessions/repetitions), variables (measurement instruments), and outcomes. A pair of reviewers (M.J.V.-G. and M.-J.E.-P.) independently evaluated all of the studies identified in the initial search, excluding articles that did not meet the selection criteria based on the title and abstract. The remaining studies were subject to a full-text evaluation using the Rayyan tool (https://www.rayyan.ai/ accessed on 1 April 2023) to identify any duplicates and ensure the inclusion criteria were met. Any discrepancies between the reviewers were resolved by a third reviewer (F.J.M.-V.) through a decision-making process based on consensus.

### 2.5. Bias Risk

Two independent investigators (M.J.V.-G. and M.-J.E.-P.) evaluated the risk of bias of each selected study using the Cochrane Collaboration Tool [40]. In the case of any uncertainties or disagreements, the authors reached a mutual agreement to resolve any discrepancies and consulted a third researcher (F.J.M.-V.) if required.

## 3. Results

### 3.1. Selection of Studies

Once the criteria for selection were applied, 584 articles were found in the different electronic databases, After duplicates were removed, abstracts were read, and the inclusion and exclusion criteria were applied, only five articles were selected [41,42,43,44,45] (Figure 1). Two of these were randomized clinical trials [44,45] and three were observational studies [41,42,43].

### 3.2. Data Extraction

#### 3.2.1. Characteristics of the Subjects

Overall, 164 participants were included: 58.3% female, aged between 21 and 75 years. The sample size ranged from 8 subjects in the research performed by Wang and co-authors [43] to 92 in the Wootenstudy [42].

Regarding the type of cancer studied in the different articles, abdominal cancers awaiting surgery [41,42,43] and breast cancer [44,45] were found (Table 2).

In 40% of the articles, the patients had completed the chemotherapy cycle and were diagnosed with cardiotoxicity caused by this treatment [44,45]. In the remaining 60%, the patients were oncology patients with abdominal cancer awaiting surgery [41,42,43]. Among the types of abdominal cancer addressed in the studies were pancreatic, colon, esophageal, rectal, retroperitoneal, small intestine, jejunum, and gall bladder cancer.

#### 3.2.2. Characteristics of the Interventions

The main characteristics, interventions, and results of the articles analyzed are shown in Table 3. Blood flow restriction was applied with exercise in all studies [41,42,43,44,45]. Regarding the type of exercise, in a few of the studies, the subject alternated every other day with BRF resistance exercises or walking [41,42,43]. In other studies, aerobic exercise with high-intensity interval training was used in combination with BFR or moderate-intensity continuous training [44,45] and nutritional supplements.

In regards to the placement of the BFR cuffs, during upper body resistance exercises, the bands were placed only on the upper arms [41,42,43], while if lower body resistance exercises or aerobic exercises such as walking on a treadmill were performed, the bands were applied to the upper part of the thighs [41,42,43,44,45].

In the case of resistance exercises, upper limb band pressures ranged from 150 mmHg to 200 mmHg and for the lower limb, they ranged from 250 to 300 mmHg, depending on the size of the arm or thigh circumference band, respectively [41,42,43]. On the other hand, for aerobic exercises, in the moderate-intensity continuous way, it was 122%, and in the high-intensity intermittent exercise, it was 52% of the optimal standard unit of KAATSU [44,45].

As for the brand of devices used for the application of the BFR, there were KAATSU, Delfi and Owens Recovery Science [44,45], and BStrong, Park City UT [41,42,43].

The strength training consisted of three sets of 20–30 repetitions with a one-minute break between sets, and the session treatment time for this therapy was 45 min [41,42,43]. Regarding the heart rate used during aerobic exercise, it ranged from 60 to 72% of the maximum heart rate [44,45].

Concerning the treatment time followed in the studies of our review, it ranged from 4 [41,42,43] to 12 weeks [44,45]. In regard to the frequency of training on a weekly basis, it ranged from 3 [44,45] to 5 or 6 days per week [41,42,43].

In some of the studies, supplements were also administered to the patients [41,42,43], and in one of them, an application was used to guide the subject during the treatment [43].

The sports nutrition supplements were whey protein, creatine monohydrate [34], and L-citrulline [41,42,43].

In three of the articles [41,42,43], intervention was used as prehabilitation before surgery. However, in two of the trials, the results were measured at the end of the treatment, before surgery [41,43], and in the other one [42], the results were measured after surgery.

In more than half of the studies [41,42,43], participants performed exercises at home following video guidance, having received specialized instruction beforehand.

In terms of the variables and measurement tools utilized, they were anthropometric indices such as weight [45], body mass index [45], waist-to-hip ratio [45], skeletal muscle mass [45] or percentage body fat [45]. All of them were measured using a body composition analyzer [45], although weight in one of the articles was measured using a balance scale [41]. Some other factors related to the aforementioned ones that were also evaluated and analyzed included caloric intake measured with the Nutrition Data System for Research [41], as well as body composition assessed via dual-energy X-ray absorptiometry [41].

Other parameters studied were quality of life assessed using the Quality of Life Questionnaire (IHF-QoL) [44], with SF-12 questionnaire [41] or with the scale health-related quality of life [43], satisfaction through surveys [43], frailty status through surveys [43], across Edmonton Frail Scale [41] or short physical performance battery [41], anxiety level [43], upper body strength with handgrip dynamometry [41], lower body strength through the 5-repetition chair stand test [41], physical function using Timed Up and Go or SPPB (short physical performance battery) [41], functional capacity by means of the six-minute walk test [41], and risk of falls [41] through Falls efficacy scale international.

Other health-related variables related to hospital stay after surgery were also stu-died: length of hospital stay [41], postoperative complications [41], readmission rate [41], and mortality at 90 days post-surgery [41].

In terms of outcomes, there were enhancements in quality of life [44], frailty [41], functional capacity [41], physical function [41], lower body strength [41], physical component score of quality of life [41], total body and appendicular lean mass, total body fat mass and trunk fat mass [41], postoperative complications [42] and length stay [42], mean steps per day on the fifth day after the operation [42] and average kcals on postoperative day 5 [42], satisfaction with the application integrating BFR [43], weight, body mass index, final score of the body analyzer, body fat, age-appropriate body, muscle tissue, waist-to-hip ratio, and basal metabolic rate [45]. The program with BFR was not related to the incidence of serious complications [42] or the readmission rate [42].

There were no notable modifications observed in hand grip strength [41], fear of falling [41], the mental component summary of quality of life [41], spontaneous physical activity in the first four days after surgery [42], in mean steps per day [42] or mean daily calories [42].

#### 3.2.3. Methodological Quality Assessment and Risk of Bias

Figure 2 and Table 4 display the outcomes of the quality evaluation for the various studies. Figure 2 exhibits the methodological excellence of the clinical trials, while Table 4 illustrates the methodological excellence of the observational studies. Regarding the observational studies, 74.2% of the STROBE Statement recommendations were fulfilled, indicating a relatively high level of compliance with the reporting guidelines. In terms of bias risk according to ROB 2.0, Adimi et al. [44] obtained the lowest risk. The domain with the lowest risk was missing outcome data (Figure 3). The domains with the highest risk were measurement of the outcome, deviations from intended interventions, and the randomization process.

## 4. Discussion

A systematic review was conducted to summarize the scientific evidence on the utilization of BFR as a therapeutic intervention for cancer survivors. Overall, the results were positive for BFR combined with exercise improving analyzed variables in all studies, except for a few exceptions [41,42]. For example, the studies found improvements in quality of life, frailty, functional capacity, and physical function [41,42,43,44], among others. Moreover, BFR was not related to serious complications or readmission rates [42].

In this section, the discussion of the topic will be approached in different sections to explore in detail various aspects related to the use of Blood Flow Restriction (BFR) in cancer patients. Below, a description of each section will be provided, and their respective key points will be analyzed.

### 4.1. The Potential of BFR in Cancer Treatment

Although many studies have analyzed the effects of exercise on cancer [47,48,49,50], the number of articles specifically examining the effects of BFR combined with exercise on cancer at any age is limited. In fact, our comprehensive literature review identified only five articles that have explored this specific topic. Further studies would be advisable, as the literature shows that it could have important effects on tumors. Cancer is linked to the presence of hypoxic areas resulting from uncontrolled cellular proliferation. This pathological hypoxia triggers several molecular signaling pathways that promote cell survival, similar to the physiological response that occurs when exposed to high altitudes, using artificial hypoxia devices, or implementing vascular occlusion of the limbs. The clinical significance of “tumor hypoxia” has increased due to its crucial involvement in both tumor progression and treatment resistance. Nonetheless, the capacity to manipulate this pathway through physical activity and systemic hypoxia-mediated signaling pathways can offer significant therapeutic opportunities that merit further exploration [51].

Exercise is a potential way to regulate tumor growth while enhancing the body’s response to cancer treatments, and a recent study has shown that engaging in leisure-time physical activity is associated with a reduced risk of developing 13 different types of cancer [52].

Additionally, the impact of systemic hypoxia on cytokines that play a critical role in tumor growth has been well-documented [53]. For instance, IL-6 in liver cancer and SPARC in colorectal cancer have been found to be affected [54]. Given the positive physiological benefits associated with exposure to systemic hypoxia, there is a plethora of new possibilities for improving prognosis and quality of life in individuals with digestive cancer [53].

### 4.2. Muscular Adaptation and Sarcopenia

Muscular adaptation due to BFR training has been attributed to the greater accumulation of metabolites, additional muscle fiber recruitment, and the resultant muscle protein synthesis [55]. Conversely, in research conducted on individuals diagnosed with breast cancer, sarcopenia was shown to be a risk factor for mortality in women with early-stage breast cancer [56]. Therefore, it is crucial to conduct further research on targeted interventions to treat sarcopenia, which could provide evidence to help reduce mortality rates among breast cancer patients [51]. Positive results have been found in older adults using BFR exercise [57].

Furthermore, training with partial blood flow restriction would be a useful tool to intervene in cancer-associated sarcopenia, constituting an alternative to induce muscle strength gain, with the reduced risks of high-intensity training [57]. It is an inexpensive and easy-to-implement technique that should be borne in mind, especially as oncology patients find it difficult to engage in physical exercise programs [58].

Systematic reviews underline the potential benefits of exercise for cancer patients [12,59,60]. Regarding the variables of influence, fatigue is one of them. In a systematic review published in 2021 [61], it was found that exercise could significantly reduce cancer-related fatigue in adults. However, none of the articles in our review analyzed this important variable. Precisely, in our opinion, this is an important factor to consider because the use of BFR therapy could potentially improve strength and muscle hypertrophy and reduce the number of repetitions required during exercise in healthy adults [16], as observed in other studies involving clinical populations [62]. This could also be beneficial for cancer patients, as cancer-related fatigue is one of their primary concerns.

### 4.3. BFR Exercise Interventions and Outcomes

In regards to BFR exercise interventions and outcomes, they show high heterogeneity, which makes it difficult to conduct meta-analyses.

Regarding the type of exercise that accompanies BFR, the study by Adimi et al. suggests that intense aerobic exercise is most beneficial for breast cancer patients who developed cardiotoxicity after chemotherapy treatment. The authors suggest that the molecular mechanisms explaining this phenomenon should be further studied, including the measurement of physiological factors such as endorphins, lactate, cortisol, testosterone, brain-derived neurotrophic factor, and other genes related to the process of neurogenesis [44].

The ACSM attempted to establish precise exercise recommendations based on cancer type, treatment, or location in 2010 [12] and 2018 [63]. The ACSM acknowledged that the existing literature was still inadequate to provide more detailed exercise guidelines for cancer survivors, a fact confirmed in a systematic review conducted in 2018, which studied high-intensity exercise in cancer patients. Improvements were found in cardiovascular capacity, strength, body mass, and quality of life. They also noted that high-intensity e-xercise can be a useful modality for improving health outcomes since it requires less time to perform it [64].

Regarding the other type of exercise used in the articles of this manuscript, strength exercise, the ACSM suggests that moderate to high weights are beneficial for improving strength and muscular endurance. However, high muscular tension exercises may not be feasible for clinical populations such as cancer survivors. Research has demonstrated that using low loads with BFR, typically around 20–30% of an individual’s one-repetition maximum (1RM), can yield similar muscle growth and strength improvements as traditional high-load training programs [65]. BFR can be implemented using a percentage of the 1RM, exercises with elastic bands, or circuit training, with loads ranging from 20% to 50% [57]. This contrasts with studies utilizing body weight and light resistance exercises, although solely using body weight can eliminate barriers associated with equipment and facility access [66].

A meta-analysis conducted by Perera et al. in 2022 revealed that BFR training enhances muscular endurance and muscle growth. Comparing low-intensity blood flow restriction training (LI-BFR) with high-intensity resistance training (HIRT), HIRT was more effective in promoting muscle hypertrophy and strength. Nonetheless, LI-BFR surpassed a similar low-intensity protocol, rendering BFR a viable option for individuals unable to manage the high loads associated with HIRT [67].

### 4.4. Frequency and Variables Studied

In terms of frequency per week, the review by Loenneke et al. found that benefits were better with two to three times weekly compared to performing the exercises four or five times per week in healthy people [16]. According to the ACSM guidelines [7], it is recommended to engage in at least 150 min of moderate exercise or 75 min of vigorous exercise per week. Exercise interventions can improve cancer-related health outcomes such as physical function, fatigue, anxiety, depressive symptoms, and health-related quality of life; nevertheless, the optimal dose of exercise for cancer survivors is unknown, and more research is needed to determine the optimal dose of exercise for cancer survivors [68]. In our review, exercise was done three to five times a week. However, when considering cancer patients, it is crucial to take an individualized approach based on their overall condition. In a cross-sectional study involving 392 people with cancer, approximately 37% of participants chose to exercise twice per week, while an additional 30% opted for three exercise sessions per week [68].

It should also be noted that in some of the studies, dietary supplements have been used during the physical exercise program. Cancer patients often experience muscle wasting and decreased physical function, leading to decreased quality of life [69]. As a result, many cancer patients turn to nutritional supplements, including those aimed at enhancing exercise performance, to help maintain muscle mass and improve physical function [70].

Secondly, the variables studied also vary widely. In terms of strength, only one of the articles analyzed the strength of the lower body [41]. However, in BFR therapy, using resistance at 20% 1RM can lead to improvements in muscular strength and power that are typically only observed with exercise at 80% of an individual’s 1RM. This makes BFR training with low loads a potentially beneficial option for patients who are unable to handle heavy mechanical loads [71]. Previous studies have demonstrated that with BFR training, individuals obtain significant strength gains, improved muscular endurance, and muscle hypertrophy [72,73]. As mentioned above, there are studies that indicate muscle hypertrophy and strength gains in healthy adults and elderly patients with sarcopenia; therefore, we believe that it would be interesting to study this variable further in oncology patients, together with another important variable such as fatigue, as the BFR could be a tool to improve strength without fatiguing the patient.

According to the American Cancer Society, one of the most common symptoms of cancer and its treatments is fatigue [74]. Patients undergoing chemotherapy often experience a range of symptoms, with fatigue being one of the most frequent and burdensome side effects. This results in impaired or reduced physical activity. While most side effects are specific to certain drugs, fatigue is commonly associated not only with most antineoplastic drugs but also with the disease itself [75].

Additionally, BFR training activates type II fast-twitch muscle fibers even at lower loads, which typically require greater intensity to activate. This explains the increased muscle hypertrophy observed in low-load BFR training compared to similar low-load exercise without BFR. Overall, the evidence provides valuable insights for guiding future research to optimize rehabilitation strategies [76].

We were also surprised to find that one of the variables analyzed in this review was hand grip strength, which is not typically examined in studies on blood flow restriction training. It is worth noting that reduced vascular function and impaired blood flow regulation during exercise can lead to a decrease in exercise capacity. This reduction in exercise capacity is negatively correlated with mortality rates in healthy individuals and may contribute to the development and persistence of cancer-related fatigue [77].

Many of the variables have been studied in the articles of this review including weight, body mass index, body analyzer score, body fat, age-appropriate body, muscle tissue, waist-to-hip ratio, and basal metabolic rate and body composition (lean mass, fat mass, trunk fat mass). Positive results have been found in all of them. This is similar to what was found in Tamakarada’s study, which suggested a potential anabolic effect of resistance training with BFR by producing a significant increase in plasma growth hormone levels [78]. However, Wooten et al. did not find any alterations in blood markers associated with protein synthesis and degradation, such as myostatin, follistatin, and growth hormone [41], and in the systematic review by Cheema et al., no significant differences were found regarding body mass composition in breast cancer survivor patients [79].

The prognostic value of assessing muscle composition, particularly intra-muscular adipose tissue, in certain types of cancer warrants further investigation, including its impact on chemotherapy toxicity and survival rates [80]. Preventing weight gain during early adulthood is crucial to avoid premature deaths, particularly in terms of controlling fat mass, which can negatively impact health and increase the risk of mortality later in life [81]. BFR training has been shown to significantly decrease body composition and body fat percentage in older women [82]. In addition, myopenia has been linked to extended hospital stay and is a significant independent prognostic factor for both disease-free and overall survival.

### 4.5. Safety Considerations

In connection with the safe use of exercise in people with cancer, it can be recommended regardless of the type of cancer. It promotes significant improvements in clinical, functional, and even survival outcomes. Generally, it is safe, but individuals should undergo screening tests and take appropriate precautionary measures [4]. If we analyze exercise with BFR training, it is a technique that can offer significant benefits in athletic performance, but caution is necessary due to potential risks. Studies have reported a low risk of adverse effects, including temporary paresthesias, bruising, and muscle soreness, but serious adverse events like rhabdomyolysis, prolonged pain, and syncopal events may occur with inappropriate usage or overexertion. The most frequently reported adverse effect is subcutaneous hemorrhaging, which occurred in 13.1% of cases, while rare cases (<0.06%) have reported more serious complications like venous thrombosis, pulmonary embolism, rhabdomyolysis, and worsening of ischemic heart disease [83]. Despite this, there is no evidence supporting an increased risk of blood clots with BFR, and it may even offer a protective effect against thromboembolic events [76].

Most studies have reported a low risk of adverse effects associated with BFR training, including transient perceptual responses and potential risks related to hemodynamics, vascular function, and thrombosis. However, these risks can be minimized with appropriate application and monitoring. A national survey of more than 12,000 Japanese individuals found no significant side effects associated with BFR training, including no cases of pulmonary embolism, cerebral hemorrhage, or venous thrombosis. Additionally, studies have shown that BFR training does not significantly impact coagulation and inflammatory responses. These findings suggest that BFR training is safe, but caution should be exercised when prescribing and implementing BFR training, particularly in individuals with underlying health conditions [57].

Also, BFR training is safe for individuals with various medical conditions, including frail individuals such as patients in the intensive care unit [84]. However, long-term studies are necessary to confirm the safety of BFR training for patients with chronic diseases who have been diagnosed with muscle wasting, such as cancer patients [14].

Regarding the effects of BFR on neuromuscular function, it has been found that, during the initial phase of exercise, BFR intensified the onset of muscle fatigue largely because of a significant decline in contractile ability. Nevertheless, the effect of BFR on muscle fatigue decreased after a 2-min reperfusion period, indicating that BFR has a potent but temporary effect on neuromuscular function [85].

In a comprehensive review that examined the impacts of strength exercise on breast cancer, it was observed that it did not increase the risk of lymphedema. However, it was not studied whether it was accompanied by BFR. In the articles included in the review, three of them involved breast cancer patients, but no adverse effects were reported regarding lymphedema [79].

Although surgery-related complications, new health issues, medication reactions, premature discharge, and failure to thrive are all potential reasons for hospital readmissions, there may be instances where preventable readmissions occur as a result of using BFR. One study reported two deaths in the group with BFR compared to none in the control group, which the authors explained by the advanced stage of cancer and frailty, along with a higher predictive risk score and slightly older age [42].

In conclusion, BFR training can offer significant benefits, but caution is necessary to minimize potential risks. When implemented within consensus guidelines, BFR does not seem to increase the risk of adverse events more than standard exercise modalities. However, appropriate methodology and the evaluation of candidates are necessary to minimize the risks of serious adverse events, and long-term studies are necessary to confirm the safety of BFR training for individuals with underlying health conditions [86].

For all the above reasons, when applying blood flow restriction, it is crucial to take into account the individual patient, occlusion pressure, cuff width, and cuff size [87]. Thigh circumference is an important predictor of BFR pressure: with larger limbs, higher pressure is required. Studies using the same exercise protocol and BFR application in men and women may yield different results [88]. Using arbitrary pressures for BFR exercise can increase cardiovascular demand, leading to adverse events such as internal bleeding or stroke. Higher cuff pressures also increase the risk of nerve injury and discomfort, which can affect adherence and the enjoyment of the exercise. On the other hand, inadequate occlusive pressure may hinder adaptations. Therefore, using arbitrary pressures may pose the greatest risk for safety and efficacy, despite being the most common approach to determine occlusion pressure [89].

The inconsistent equipment used to induce BFR is a constant limitation. The narrower cuffs require higher pressures to completely occlude blood flow in the limbs compared to elastic cuffs or nylon cuffs. According to some research, it would be appropriate to work with a PR between 50 and 60% of the value required to achieve Limb Occlusion Pressure (LOP) [57]. However, it is common for researchers not to communicate this information, making it impossible to reproduce the results if only the pressure used is reported without indicating the percentage of arterial occlusion pressure (AOP) or limb occlusion pressure (LOP) or the equipment used. Nevertheless, this problem can be mitigated with the use of individualized pressures, as long as the LOP, AOP or a percentage of these values [90] uses the same cuff employed during the exercise. Efforts are made to ensure that BFR exercises are comfortable for the patient. Research has shown that using cuffs with intermittent inflation patterns is less uncomfortable than using cuffs with continuous inflation patterns [90].

The optimal volume, type, and dose of exercise needed to induce positive results in patients are unclear. Therefore, it is important for a doctor to conduct a proper evaluation before patients participate in this modality until more research is published [64].

### 4.6. Use the BFR in Prehabilitation

On the other hand, we would like to emphasize the use of pre-surgery exercise programs, known as prehabilitation, in cancer patients who are going to undergo surgery.

Prehabilition is safe and feasible for cancer patients. These programs can enhance functional capacity after surgery, as indicated by improved 6-min walk test results. Surgery is associated with adverse effects, such as reduced fitness, high complication rates, emotional distress, and poor quality of life [91]. Optimizing functional capacity before surgery is crucial, especially for patients with poor survival rates. However, compliance with prehabilitation may be a challenge, particularly in patients with severe diseases, high comorbidities, and neoadjuvant therapy. Further investigation is needed to assess the effects of prehabilitation on patients with low physical fitness, poorer prognosis, and different cancer types.

Traditional prehabilitation programs have been centered in hospitals or healthcare facilities, which require frequent supervised visits. This demand can be challenging and impractical in terms of accessibility, expense, and time, often leading to lower adherence rates [92]. However, a home-based multimodal prehabilitation program has been shown in some of the articles in our review to be feasible and enjoyable for participants, with excellent adherence rates. Furthermore, the prehabilitation program currently in place has received positive feedback from participants and their caregivers/family members through personal communication. Additionally, participants have reported feeling comfortable, confident, and at ease when learning how to use BFR bands. This suggests that a larger study could be implemented successfully for individuals who prefer light-intensity exercise in a home-based setting to improve physical readiness before surgery [41,42].This is a significant departure from previous unsupervised multimodal prehabilitation programs, which reported low (45%) and moderate (78%) compliance rates to their interventions [93,94].

A prehabilitation program can lead to significant improvements in postoperative functional exercise capacity [93].

BFR training has potential for faster recovery in post-operative rehabilitation, but the risk of venous thromboembolism is a concern, especially in high-risk patients [67]. Prehabilitation may be a better option for promoting rehabilitation in clinical populations. Further research is needed to determine the risk–benefit ratio of perioperative BFR protocols in clinical populations.

### 4.7. Limitations and Future Directions

The present analysis has certain limitations due to the small number of articles included and the limited availability of literature, despite the exhaustive literature search in 15 databases. However, positive outcomes of blood flow restriction in cancer survivors are observed. The lack of consistency in the studies made some analyses impossible, possibly due to differences in exercise protocols, duration and number of sessions, length of treatment, and the procedure performed. The studies also failed to take into account possible co-morbidities that could affect outcomes, BFR application, and methods used to evaluate the same variable. A third limitation is the absence of long-term follow-up to assess the effects of BFR and compare it to other interventions. Additionally, participants in all studies were not blinded, and therapists were unable to mask the treatment. The methodological quality was high in observational trials and with moderate risk according to the ROB 2 tool for ECAs. In addition, two of the five studies [44,45] used the same sample of patients, although they are considered different publications because the variables studied were different. The other three articles [41,42,43] used the same intervention group to conduct different types of studies. It should also be noted that very few types of cancer have been studied, and the different stages of the diagnosis and treatment process at which they were found have not been taken into account. Furthermore, medication and food supplementation provided in some protocols could influence the results.

Due to the high prevalence and incidence of cancer [95], it is essential to explore strategies that can improve the quality of life of cancer survivors without worsening their symptoms. The results of this review pave the way for conducting further clinical trials with a larger number of subjects with different types of cancer to confirm the benefits found here and better analyze this powerful tool in this clinical population, considering different stages cancer, and incorporating standardized protocols.

## 5. Conclusions

The findings of this research suggest that BFR may also be a promising intervention for cancer patients and survivors. It can be used in the pre-habilitation period in cancer patients awaiting surgery. The incidence of adverse events related to exercise and BFR was low, but given the limited number of studies found and significant variation, solid conclusions cannot be drawn. It is essential that therapists determine the most appropriate protocol to implement BFR. In order to improve the understanding and effectiveness, future research should concentrate on tailoring its application to individual patients, optimizing load progression, ensuring long-term follow-up, and enhancing the methodological rigor of studies, such as implementing sample blinding.

## Figures and Tables

**Figure 1 healthcare-11-02062-f001:**
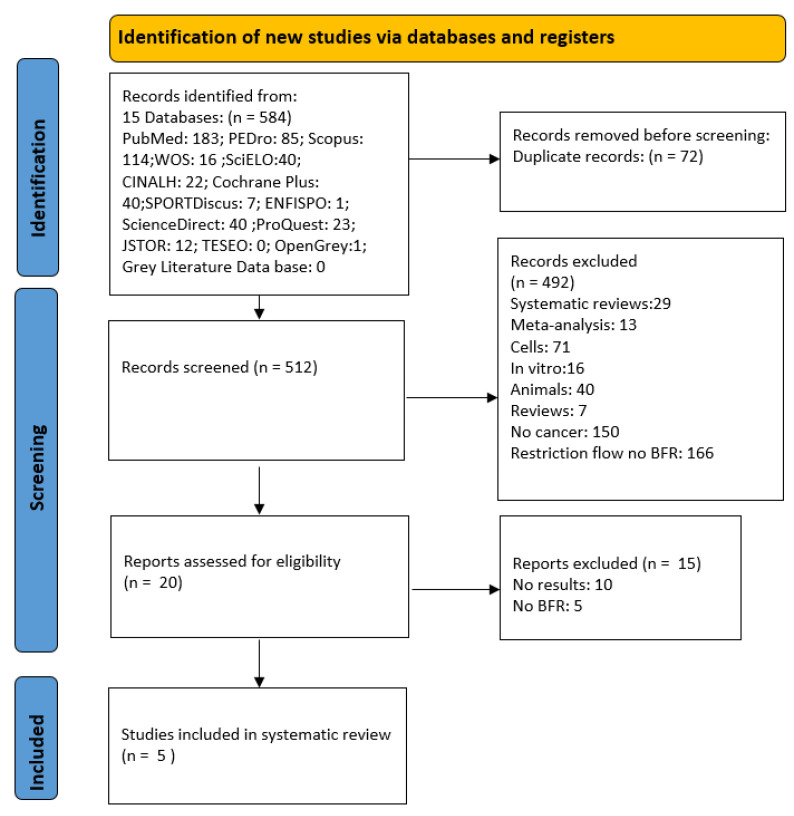
PRISMA 2020 flow diagram.

**Figure 2 healthcare-11-02062-f002:**
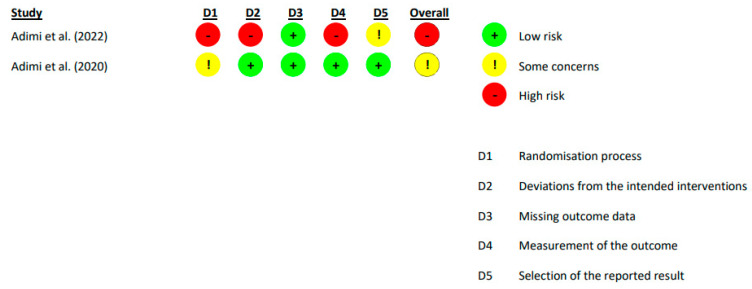
Risk of bias of the studies included in the systematic review [44,45].

**Figure 3 healthcare-11-02062-f003:**
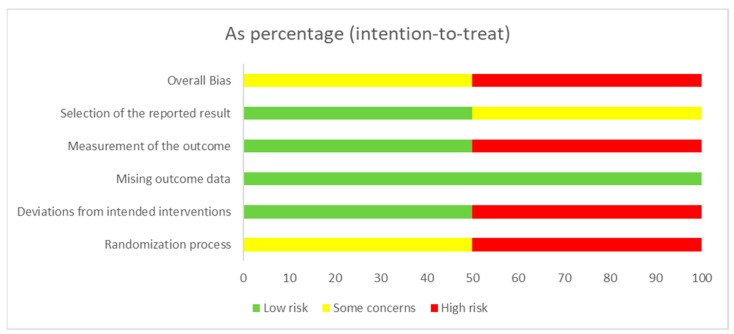
General risk of bias. Each category is presented as a percentage.

**Table 1 healthcare-11-02062-t001:** Search strategy followed in the different databases.

Databases	Search Strategy
PubMed	(cancer* OR neoplasm* OR malignan* OR carcinoma OR tumor OR tumor OR oncolog*) AND (blood flow restriction therapy OR Limb occlusion pressure OR blood flow restriction OR Blood flow restriction exercise OR BFR exercise OR KAATSU training OR limb occlusion OR blood flow restrict* OR vascular restrict* OR KAATSU)
PEDro	blood flow restriction
Scopus	(cancer* OR neoplasm* OR malignan* OR carcinoma OR tumor OR tumor OR oncolog*) AND (Limb occlusion pressure OR “blood flow restriction” OR “Blood flow restriction exercise” OR “Blood flow restriction therapy” OR “BFR exercise” OR “KAATSU training” OR “limb occlusion” OR “blood flow restrict*” OR KAATSU)
WOS	(cancer* OR neoplasm* OR malignan* OR carcinoma OR tumor OR tumor OR oncolog*) AND (Limb occlusion pressure OR blood flow restriction OR Blood flow restriction exercise OR BFR exercise OR KAATSU training OR limb occlusion OR blood flow restrict* OR vascular restrict* OR KAATSU)
SciELO	“blood flow restriction” AND oncolog* OR cancer OR neoplasm In all indexes
CINAHL	(cancer* OR neoplasm* OR malignan* OR carcinoma OR tumor OR tumor OR oncolog*) AND (Limb occlusion pressure OR blood flow restriction OR Blood flow restriction exercise OR BFR exercise OR KAATSU training OR limb occlusion OR blood flow restrict* OR vascular restrict* OR KAATSU)
Cochrane Plus	blood flow restriction AND cancer in title, abstract, keywords
SPORTDiscus	(cancer* OR neoplasm* OR malignan* OR carcinoma OR tumor OR tumor OR oncolog*) AND (Limb occlusion pressure OR “blood flow restriction” OR “Blood flow restriction exercise” OR “BFR exercise” OR “KAATSU training” OR “limb occlusion” OR “blood flow restrict*” OR KAATSU)
ENFISPO	(Limb occlusion pressure OR blood flow restriction OR Blood flow restriction exercise OR BFR exercise OR KAATSU training OR limb occlusion OR blood flow restrict* OR vascular restrict* OR KAATSU)
ScienceDirect	“Blood Flow Restriction Therapy”
ProQuestResearch Library	abstract (blood flow and exercise) AND abstract (Cancer)
Journal Storage	“blood flow restriction”
TESEO	“blood flow restriction”
OpenGrey	“blood flow restriction”
Grey Literature Database	“blood flow restriction”

For all databases included in the review, the search was conducted without a date limit and it went until May 2023.

**Table 2 healthcare-11-02062-t002:** Participant´s Characteristics in Terms of Cancer Type.

Author/Country	Type of Cancer	Stage of the Disease	Type of Treatment
Adimi et al. (2022) [45] Iran	Breast cancer with cardiotoxity	Not mentioned	chemotherapy
Wooten et al. (2022) [42] United States	Abdominal cancer: Pheochromocytoma, Colon with/without hepatic metastasis, Esophageal, Gall bladder, Jejunum, Pancreas, Stomach, Rectal with/without hepatic metastasis, Retroperitoneal, Small intestine with/without hepatic metastasis, Cecum, Leiomyosarcoma, Liver	CG: 2.7 ± 1.6 EC: 3.3 ± 1.0	Underwent elective cancer-related surgery and usual preoperative care
Wooten et al. (2021) [41] United States	Abdominal cancer: Pheochromocytoma/Adrenocortical, Colon or/with Hepatic Metastasis, Esophageal, Gall Bladder 1 Jejunum, Pancreas, Rectal or/with Hepatic Metastasis, Retroperitoneal, Small Intestine or/with Hepatic Metastasis	- 8.3%: stage 1 - 8.3%: stage 2 - 25%: stage 3 - 58.3%: stage 4	Surgery with different complexity: simple, intermediate or complex.
Wang et al. (2021) [43] United States	Abdominal cancer	Not mentioned	surgery
Adimi et al. (2020) [44] Iran	Breast cancer with cardiotoxicity	Not mentioned	- chemotherapy course completed - cardiac rehabilitation

**Table 3 healthcare-11-02062-t003:** Main characteristics and results of the studies.

Author/Type of Cancer	Treatment	Type of BFR	Variables/Assessment Tools	Results
Adimi et al. (2022) [45] Breast cancer	n = 20 G1: n = 5 HITT G2: n = 5 MIT G3: n = 5 HITT+BFR G4: n = 5 MIT+BFR Treadmill 3 days/week 12 weeks	Not mentioned	- weight (body analyzer) - BMI (body analyzer) - WHR (body analyzer) - % body fat (body analyzer) - SMM (body analyzer)	- BMI, WHR, % body fat, SMM improved in HITT + BFR (*p* < 0.05) - BMI: HIIT + BFR (25.5 ± 2.5) vs. MIT (24.5 ± 2.7); *p* = 0.001 - WHR: HIIT + BFR (0.8 ± 0.1) vs. MIT (0.9 ± 0.05); *p* = 0.043. - BF%: HIIT + BFR (32.5 ± 2.5) vs. MIT (30 ± 5); (*p* = 0.003). - SMM: HIIT + BFR (26 ± 2.5) vs. MIT (25 ± 3.3); *p* = 0.003)
Wooten et al. (2022) [42] Abdominal cancer	n = 92 BFR group: n = 21 Nutrition supplement + home-based exercise of low-intensity upper and lower body (followed video 45 min) + BFR resistance exercises or 15 min of walking with leg BFR bands (alternated every other day) 5–6 days/week 4 weeks CG: n = 71 no prehabilitation (prior surgery)	- BFR bands: (BStrong, Park City, UT, USA). - They were inflated to the pre-determined pressures recommended by the manufacturer	- Physical activity levels (acelerometers) in CG - length of hospital stay, - postoperative complications, - readmission rate, - mortality at 90 days, - post-surgery. (variables measured after surgery)	- Length of Hospital Stay −5.5 days (Cohen’s d); BFR group: 4.7 ± 2.1 vs. CG: 10.2 ± 1.2 days (*p* = 0.02)). - Any Complications; decreased incidence of complications (38 vs. 69%) (*p* = 0.03); 0.38 (OR) - Serious Complications: BFR group: not related to incidence of serious complications (*p* = 0.17), 1.19 (OR) - Readmission Rate: BFR group: not related to the readmission rate (*p* = 0.59), 0.78 (OR) - BFR group: 58% more steps on day 5 after surgery (*p* = 0.043). - no significant difference in mean steps per day - BRF group: 91% more kcals on average vs. CG on postoperative day number 5 (*p* = 0.049)
Wooten et al. (2021) [41] Abdominal cancer	n = 24 BFR + nutrition supplement Body weight and light resistance exercises+ BFR 3 sets (20–30 repetitions/1-min rest between sets) or 15 min of walking with leg BFR bands (alternated every other day) 5–6 days/week 4 weeks (prior surgery)	- BFR bands (BStrong) - Upper arm BFR bands were 4.5 cm in width, and leg BFR bands were 6 cm in width. - They were inflated to the pre-determined pressures according to BFR band size (18 × 30.5 cm upper arm circumference band: 150 mmHg, 30.5–44.5 cm upper arm circumference band: 200 mmHg, 44.5–60 cm thigh circumference band: 50 mmHg, and 60–78.5 cm: 300 mmHg).	- Physical function, bodily pain, general health, vitality, social functioning, mental health (SF-12) - risk of falls (FES-I) - caloric intake (NDSR) - height and body weight (balance scale) - body composition (dual energy X-ray - absorptiometry): Appendicular lean mass was calculated by adding arm and leg lean mass - upper body strength (handgrip dynamometry) - lower body strength (5-repetition chair stand test) - physical function (TUG/SPPB) - functional capacity (6MWT) - frailty (SPPB and EFS) (variables measured prior to surgery)	- Improvements in 6MWT, TUG, SPPB, 5-chair stand test, and physical component score of quality of life (*p* < 0.05). - Increase in total body mass: 0.73 ± 1.04 kg (*p* = 0.004). Standard error: ± 1.04 kg; ES: 0.70. - Appendicular lean mass: 0.42 ± 0.64 kg (*p* = 0.006), standard error: ±0.64 kg; ES: 0.66. - Total body fat mass and trunk fat mass decreased (*p* = 0.004 and *p* = 0.021). - No significant changes in hand grip strength, fear of falling, the mental component summary of quality of life. - 6 MWT: change: +49 m; standard error: ±53 m; ES: not provided; *p*: <0.01 Lower Body Muscle Strength (5-Repetition Chair Stand Test) Time to complete the test reduced significantly Baseline: 14.6 s, End of prehabilitation: 9.8 s *p*-value: 0.03. - TUG: Improved average of 0.90 s Standard Error: ±0.72 s ES: Not provided *p*: <0.01 - SPPB: decreased: Baseline: 9.6 points 4-weeks: 10.8 points *p*-value: 0.01 - SF-12 (physical component): improved by 7.5 points; standard error: ±10.2 points; ES: not provided; *p*: 0.01
Wang et al. (2021) [43] Abdominal cancer	n = 8 BFR exercise and sport nutrition supplement intervention (pro-gram integrated into a mobile app) 4 weeks	Band placement refers to the process of setting up and inflating the BFR (Blood Flow Restriction) bands on users’ arms or legs. This instructional video, created by B Strong, LLC, provides guidance on how to properly set up and use the bands for BFR training.	- satisfaction (survey includes the assessment of feelings (enjoyment, difficulty with the prehabilitation program, ease of use of the app, and information load)) - frailty status - health-related quality of life-anxiety level	- Simplicity 5 ± 0.1 - Interface Design High 4.88 ± 0.12 - Organized Information 4.63 ± 0.37 - Functionality 4.88 ± 0.12 - Overall Satisfaction 4.88 ± 0.12
Adimi et al. (2020) [44] Breast cancer	n = 20 G1: n = 5 HITT G2: n = 5 MIT G3: n = 5 HITT+BFR G4: n = 5 MIT+BFR Treadmill 3 days/week 12 weeks	- BFR in MIT: 122%. - BFR in HITT: ranged from 52% of the optimal standard unit (SKU) of KAATSU. - BFR was achieved using the KAATSU device by closing the cuff on the thighs and applying pressure based on individual reference pressure and maximum pressure measurements.	- quality of Life (IHF-QoL)	- Quality of life improved in G3 - Social Life-Disrupting Symptoms: ES: 2.523, *p* = 2.222, F: 2.25 - Daily Activity ES: 2.635; *p* = 2.222, F: 6 - Total Quality of Life Score: ES: 2.032, *p* = 2.221, F: 13.62

CG: Control group; IG: intervention group; BFR: blood flow restriction therapy; BMI: body mass index; WHR: Waist to hip ratio; SMM: Skeletal muscle mass; FES-I: Falls efficacy scale international; NDSR: Nutrition Data System for Research; 6MWT: six-minute walk test; min: minute; TUG: Timed Up and Go; SPPB: short physical performance battery; EFS: Edmonton Frail Scale; HITT: high-intensity interval training; MIT: moderate-intensity continuous training; app: application: IHF-QoL: Quality of Life Questionnaire; ES: effect size; *p*: *p*-value; F: F-test; OR: odds ratio.

**Table 4 healthcare-11-02062-t004:** Evaluation of the quality of observational studies using the STROBE Statement [46].

Analyzed Portion	Object	Wooten et al. (2021) [41]	Wooten et al. (2021) [42]	Wang et al. (2021) [43]
Title and abstract	1		×	×
I: background/rationale	2	×	×	×
I: objectives	3	×	×	×
M: study design	4			
M: setting	5	×		×
M: participants	6	×	×	×
M: variables	7	×	×	×
M: data sources/measures	8	×	×	×
M: biases	9		×	
M: study size	10			
M: quantitative variables	11	×	×	
M: statistical methods	12	×	×	
R: participants	13	×	×	×
R: descriptive data	14	×	×	×
R: outcome data	15	×	×	×
R: main results	16	×	×	×
R: other analyses	17			
D: key results	18	×	×	×
D: limitations	19	×	×	×
D: interpretation	20	×	×	×
D: generalizability	21	×	×	
D: Other information: funding	22	×	×	

I: Introduction; M: material and methods; R: results; D: discussion. ×: meets criterion

## Data Availability

Upon request, the corresponding author will make the data presented in this study available.

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
