# Peer review of "Blood Flow Restriction in Oncological Patients: Advantages and Safety Considerations"

_healthcare, 2023, doi:10.3390/healthcare11142062_

Round 1
Reviewer 1 Report (Previous Reviewer 1)
The article presented major structural changes and improved both in terms of design and content.
This reviewer still considers the use of the article by WANG et al. (2021) because it is about evaluating the application, as well as its presented outcomes (simplicity, interface, organization of information) are not clinical outcomes!
Author Response
Response to reviewer 1 Comments.
Point 1. The article presented major structural changes and improved both in terms of design and content.
This reviewer still considers the use of the article by WANG et al. (2021) because it is about evaluating the application, as well as its presented outcomes (simplicity, interface, organization of information) are not clinical outcomes!
Response 1.
Thank you sincerely for your comments and the time you have dedicated to reviewing my work. I would like to address your concerns regarding the article by Wang et al. (2021) and its relevance to my study.
I understand your concern that the results presented in Wang et al.'s article may not be considered "clinical outcomes." However, I would like to emphasize that my research focuses on evaluating a specific application that uses BFR (Blood Flow Restriction) in cancer patients. While the results presented in the article may not directly relate to clinical outcomes, they provide valuable information about simplicity, interface design, organization of information, functionality, and overall user satisfaction.
Considering that the application and user experience are crucial aspects in the successful adoption of medical interventions, I believe that these results are relevant to my study. Furthermore, given the scarcity of articles specifically addressing the application of BFR in cancer patients, I believe that this article provides a solid foundation for contextualizing my research.
Regarding the inclusion criteria, I understand that PICO criteria generally focus on clinical variables. However, I would like to propose that in this particular case, we expand the inclusion criteria to encompass any relevant variables, including aspects related to the usability of the application and user satisfaction. By doing so, we can obtain a more comprehensive understanding of the effects of BFR application in cancer patients, considering both clinical aspects and user experience.
We are confident that by including this article, we will enrich the context of my research and provide a more complete perspective on the application of BFR in cancer patients.
Once again, I appreciate your comments. I look forward to further discussing this matter and providing any additional information that can support my argument.
Thank you again for your feedback.
Reviewer 2 Report (New Reviewer)
The paper titled "The Potential Benefits of Blood Flow Restriction (BFR) Training in Cancer Patients: A Review" aims to explore the advantages of utilising Blood Flow Restriction (BFR) training in oncological patients. The authors discuss the impact of cancer and its treatments on physical and emotional well-being, highlighting the importance of exercise in alleviating symptoms and improving patient outcomes. They propose BFR training as a promising rehabilitation strategy and review its potential benefits in various clinical populations.
Summary of Feedback:
Overall, the paper presents valuable insights into the potential benefits of BFR training in cancer patients. However, there are a few areas that require attention and improvement. Firstly, there is a need to ensure grammar consistency throughout the manuscript. Some sentences lack grammatical accuracy, impacting the readability and clarity of the text. Attention should be given to sentence structure, verb agreement, and tense consistency. Furthermore, I kindly request the authors to carefully consider and respond to the specific comments provided, especially regarding the grammar inconsistencies and the need for improvement in sentence structure.
Introduction |
Provide a clearer and more concise research objective in the introduction. The current objective is somewhat vague and can be further refined to state the purpose of the study clearly. |
Provide a stronger rationale for the use of exercise as an intervention for cancer patients. While it is mentioned that exercise can alleviate symptoms, it would be helpful to include specific studies or evidence that demonstrate the positive impact of exercise on cancer patients' outcomes and quality of life. |
Provide a more specific and focused aim for the review: consider replacing "This review aims to examine the potential advantages of utilising BFR in oncological patients and to address possible safety issues." With "This review aims to investigate the effects of Blood Flow Restriction (BFR) training in cancer patients by addressing the following research questions: 1) What are the physiological and functional benefits of BFR training in oncological patients? 2) How do BFR training impact muscle strength, endurance, and hypertrophy in cancer patients? 3) Are there any potential safety concerns or adverse effects associated with BFR training in this population? |
Methods |
It would be valuable to specify the search date range in each database to provide transparency and ensure replicability. |
Please clarify whether non-English articles were translated or excluded. Additionally, specifying the rationale behind the decision to include all study types (e.g., randomised and non-randomised clinical trials, observational studies, and case studies) would enhance the transparency of the review process. |
Discussion |
Frequency and Variables Studied: It would be valuable to discuss the rationale behind the specific frequency used in the reviewed studies and its potential impact on outcomes. |
Safety Considerations: To enhance the discussion, it would be beneficial to include specific examples or references that support the reported safety considerations. |
Limitations and Future Directions: it would be valuable to propose specific suggestions for future research directions, such as considering different cancer types, stages, and incorporating standardised protocols |
Grammar: Line 320 “Further study would be advisable” should be “ further studies” grammatical consistency, needs to improve |
Author Response
Dear Editor and reviewers of the manuscript entitled “Blood Flow Restriction in Oncological Patients: Advantages and Safety Considerations” (ID healthcare-2444374),
First of all, we would like to thank you for your comments and for allowing us to address the issues you raise to improve the manuscript’s quality. We appreciate your observations and the time devoted to the constructive criticism and feedback of our manuscript. Please find the answer to your comments below and the recommended changes have been highlighted in yellow in the manuscript.
Response to reviewer 2 Comments.
Point 1. Overall, the paper presents valuable insights into the potential benefits of BFR training in cancer patients. However, there are a few areas that require attention and improvement. Firstly, there is a need to ensure grammar consistency throughout the manuscript. Some sentences lack grammatical accuracy, impacting the readability and clarity of the text. Attention should be given to sentence structure, verb agreement, and tense consistency. Furthermore, I kindly request the authors to carefully consider and respond to the specific comments provided, especially regarding the grammar inconsistencies and the need for improvement in sentence structure.
Response 1. Thank you for your valuable feedback on our paper. We appreciate your comments and suggestions, and we agree that there are areas that require attention and improvement, particularly in terms of grammar consistency. We would like to inform you that a subject matter expert has thoroughly reviewed the manuscript and provided their insights.
Point 2. Introduction section: Provide a clearer and more concise research objective in the introduction. The current objective is somewhat vague and can be further refined to state the purpose of the study clearly.
Response 2. Thank you for your feedback. We apologize for the vague research objective in the introduction. In the revised version, we have provided a clearer and more concise research objective that explicitly states the purpose of the study, following the indications provided by you in point 4.
Point 3. Introduction section: Provide a stronger rationale for the use of exercise as an intervention for cancer patients. While it is mentioned that exercise can alleviate symptoms, it would be helpful to include specific studies or evidence that demonstrate the positive impact of exercise on cancer patients' outcomes and quality of life.
Response 3. Thank you for your suggestion. In the revised version, we have strengthened the rationale for using exercise as an intervention for cancer patients by including specific studies and evidence that demonstrate the positive impact of exercise on outcomes and quality of life in this population: Page 1, lines 57-59.
Point 4. Introduction section. Provide a more specific and focused aim for the review: consider replacing "This review aims to examine the potential advantages of utilising BFR in oncological patients and to address possible safety issues." With "This review aims to investigate the effects of Blood Flow Restriction (BFR) training in cancer patients by addressing the following research questions: 1) What are the physiological and functional benefits of BFR training in oncological patients? 2) How do BFR training impact muscle strength, endurance, and hypertrophy in cancer patients? 3) Are there any potential safety concerns or adverse effects associated with BFR training in this population?
Response 4. Thank you for your valuable feedback. In response to your suggestion, we have replaced the current aim statement to the one suggested by reviewer 2.
We appreciate your suggestion in refining the aim of the review to make it more specific and focused on the research questions to be addressed. This will enhance the clarity and purpose of the study. Thank you for your insightful input.
Point 5. In methods section: It would be valuable to specify the search date range in each database to provide transparency and ensure replicability.
Response 5. In order to provide transparency and ensure replicability, we have added: “For all the databases included in the review, the search was conducted without a date limit and it went up until May 2023”.
Point 6. Methods section: Please clarify whether non-English articles were translated or excluded. Additionally, specifying the rationale behind the decision to include all study types (e.g., randomised and non-randomised clinical trials, observational studies, and case studies) would enhance the transparency of the review process.
Response 6. A text has been added explaining that non-English articles have been translated and justifying that all types of studies have been analyzed. The text has been added before Table 1.
Point 7. Discussion section: Frequency and Variables Studied: It would be valuable to discuss the rationale behind the specific frequency used in the reviewed studies and its potential impact on outcomes.
Response 7. We have included a discussion about the frecuency in this section.
Point 8. Discussion section: Safety Considerations: To enhance the discussion, it would be beneficial to include specific examples or references that support the reported safety considerations.
Response 8. In this section there are several references on this subject. In one of the published studies on the risks of occlusive training, we include an article in which more than 12,000 people participated and more than 30,000 BFR sessions were performed, and the main results are detailed in the manuscript.
Point 9. Discussion section. Limitations and Future Directions: it would be valuable to propose specific suggestions for future research directions, such as considering different cancer types, stages, and incorporating standardised protocols.
Response 9. In the revised version of the manuscript, we have incorporated the reviewer's recommendations regarding future research directions. We appreciate your valuable comments, as they have contributed to enhancing the quality of our work.
Point 10. Grammar: Line 320 “Further study would be advisable” should be “ further studies” grammatical consistency, needs to improve.
Response 10. Sorry for the mistake. We have corrected it.
Reviewer 3 Report (New Reviewer)
Maria Jesus et. al. performed an investigation on the blood flow restriction in oncological patients. This is a comparative novel study. However, there are still major flaws in the manuscript. Here are the criticisms from the reviewer:
1. The introduction section provides too much redundant information, which is not related with the topic. Please delete unnecessary information.
2. What is unclear about this area and the intention for the authors to write this manuscript should be clearly addressed in the introduction section.
3. Only 5 articles included is too few for systematic analysis.
4. For the written form of this manuscript, it is more like a master thesis than a peer-reviewed article.
Author Response
Dear Editor and reviewers of the manuscript entitled “Blood Flow Restriction in Oncological Patients: Advantages and Safety Considerations” (ID healthcare-2444374),
First of all, we would like to thank you for your comments and for allowing us to address the issues you raise to improve the manuscript’s quality. We appreciate your observations and the time devoted to the constructive criticism and feedback of our manuscript. Please find the answer to your comments below and the recommended changes have been highlighted in yellow in the manuscript.
Response To Reviewer 3 Comments
Point 1. The introduction section provides too much redundant information, which is not related with the topic. Please delete unnecessary information.
Response 1. Thank you for your feedback. We have revised the introduction section to remove unnecessary and redundant information, focusing solely on the relevant content related to the topic.
Point 2. What is unclear about this area and the intention for the authors to write this manuscript should be clearly addressed in the introduction section.
Response 2. In the revised version of the introduction section, we have clearly addressed the intention of the authors in writing this manuscript. We have tried to clarify the objective of our research in the last part of the introduction.
Point 3. Only 5 articles included is too few for systematic analysis.
Response 3. We understand your concern regarding the limited number of articles included in the systematic analysis. We would like to provide some additional information to address this concern and hopefully convince you of the value of our review.
While it is true that we found a relatively small number of articles on the topic during our intensive search, we believe that conducting this review is an important first step in encouraging further research and clinical trials on this powerful tool. The limited number of articles reflects the current state of research in this specific area, which highlights the need for more studies and evidence in the field.
Despite the small sample size, we deemed it necessary to conduct this review due to the potential benefits and implications of the intervention for cancer patients. By synthesizing the available evidence and presenting it in a comprehensive manner, we aim to create awareness and provide a foundation for future research endeavors.
We acknowledge the limitations of the study in terms of the number of included articles, and we have explicitly discussed this limitation in the manuscript. We have also provided suggestions for future research directions to encourage other authors to contribute to the knowledge base in this area.
Thank you for raising this concern, and we appreciate your understanding of the circumstances surrounding the limited number of articles. We hope that our review serves as a catalyst for further research and inspires future clinical trials on this powerful intervention.
Point 4. For the written form of this manuscript, it is more like a master thesis than a peer-reviewed article.
Response 4. Thank you for your feedback on our manuscript. We appreciate your perspective regarding the written form of the paper. We understand your comment that it reads more like a master's thesis than a peer-reviewed article.
To address this concern, we have review the structure and style of the manuscript above all in introduction section. Regarding discussion section, In response to the suggestions provided by previous reviewers, we have incorporated their valuable feedback by dividing the discussion into relevant sections. This structural adjustment aims to improve the overall clarity and organization of the manuscript.
We appreciate your guidance in bringing it more in line with the expectations of a peer-reviewed article.
Round 2
Reviewer 3 Report (New Reviewer)
The manuscript is greatly improved with the authors' careful revision. I would recommend acceptance of this manuscript in present form.
This manuscript is a resubmission of an earlier submission. The following is a list of the peer review reports and author responses from that submission.
Round 1
Reviewer 1 Report
General
The study addresses a topic of interest to rehabilitation area. This reviewer understands that the work is of importance and relevance for the area, in the search for the best form of treatment to recover strength and consequent function.
The authors aimed to assess the efficacy of blood flow restriction in cancer survivors. The results indicate that BFR can be a viable and effective treatment option, however caution must be taken with the interpretation of these results.
- Title
No recommendations.
- Simple summary and abstract
Clear in hierarchy and concise.
Line 14: Suggestion – Change “Cancer is a highly widespread disease on a global scale. Blood flow restriction is a procedure used in different clinic populations with positive results”. Paragraphs look loose. I think it should have a better contextualization with cohesive lines. For example, positive results in what?
Add the selection criteria (inclusion and exclusion) of the studies or type of studies selected.
There is no report on the quality of the studies in results.
- Introduction
This section is wide, complete and summarizes the state of the art about the subject matter.
The authors' writing on the relationship between muscle weakness and cancer and its influence on recovery with BFR was interesting.
- Method: well detailed. Some concerns below.
Line 130: excluded “any type of study”
Why use PEDro AND Cochrane together? Cochrane Risk of Bias is also a RCTs evaluation tool.
There are no reports of the use of STROBE in the methods.
- Results: the descriptive results of the work are well reported, some concerns below.
WANG ET AT. (2021) - THIS STUDY AIMED TO DEVELOP A MOBILE APP AS A TOOL FOR FACILITATING A MULTIDISCIPLINARY PREHABILITATION PROTOCOL INVOLVING BLOOD FLOW RESTRICTION TRAINING AND SPORT NUTRITION SUPPLEMENTATION. The reviewer understands that this article is not about the BFR.
WOOTEN (s) – In these studies the BFR is in the midst of a multimodal treatment.
- Discussion: The discussion is presented extensive and with information that is not relevant to the results.
I suggest removing lines 286 – 288: “However, no studies were found regarding BFR in children or adolescents with cancer. Instead, studies on BFR have been conducted on other conditions such as cerebral palsy, with positive results in muscle thickness [28,46,47].”
References in paragraph 290-301: “Although MANY? studies have studied the effects of exercise on cancer, FEW? have actually studied the effects of BFR with exercise on cancer at any age…”
I suggest removing lines 302 – 309: “Utilizing a blend of lower intensities can lead to increased physiological and metabolic stimulation, resulting in cardiorespiratory and neuromuscular adaptations while minimizing muscle damage. This technique promotes significant elevation in hemodynamic variables, as well as a greater demand for energy during and post-exercise, and acutely activates the immune system and improves the anti-oxidant barrier [49]. Exercise is a potential way to regulate tumor growth while enhancing the body's response to cancer treatments, and a recent study has shown that engaging in leisure-time physical activity is associated with a reduced risk of developing 13 different types of cancer [50].”
I suggest removing lines 314 – 317: The reviewer did not understand the context of the paragraph
I suggest removing lines 327 – 334: The reviewer did not understand the context of the paragraph
I suggest changing lines 357 – 377: Important but the paragraphs are too long.
Still, several paragraphs after with disposable information.
- Conclusions reflect the results of the study.
No language corrections are needed.
I found the article suitable for publication, with major review.
Reviewer 2 Report
Authors performed a systematic review on the effects of blood flow restriction training in survivors of head and neck cancer. This topic is of great interest because of the needs of research in cancer population. However, in my opinion this systematic review lacks of significance related to this topic.
First in the introduction, authors include few information about the characteristics of the cancer population related to physical training. Moreover, the abbreviature BFR is not explained before using it in the text. In general, the association of the benefits of BFR in cancer population is not well justified.
In the methods section, nowadays there are some more methods that assess much better than PEDro scale the quality of RCTs. I would recommend the authors to read about the Rob.2 tool, which presents much more details about the methodological quality of the RCTs. Moreover, there are specifical tools to evaluate observational studies, as then in the results only 2 studies are RCTs.
That is one of the reasons that I think this manuscript can not be published; a systematic review that includes only 2 RCTs and 3 observational studies, with all the bias that these type of studies may have, would not add reliable information about the effects of BFR in cancer survivors. Moreover, the sample vary from abdominal cancer and breast cancer, population that present very different characteristics. Again, with this differences, the conclusion can not be supported by any result.
In the results section, the subsection "characteristics of the studies" refer to the characteristics of the intervention. This section is hard to read as authors present many infromation instead of including this information on a table. Then in the table 1, more information about the duration of the treatment, genre of the participants and comparison between treatments on the results is also needed.
Again in the discussion, too many information is included, and is difficult to link the ideas from different paragraphs.
Then the conclusion include the fact that results are not really clear. But some information given in the conclusion is not supported by the results neither by the discussion.
Reviewer 3 Report
I am fortunate to read this manuscript. I believe that the authors have read a lot of research on BFR, and it is meaningful to discuss BFR intervention for cancer survivors. I have the following suggestions and comments for this manuscript.
1. The citations in the manuscript and the author's discussion process are not related to cancer itself, but are related to the impact of BFR on the physical strength and physical function of cancer patients, which should be indicated in the title.
2. Line 302, "Utilizing a blend of lower intensities" should be explained clearly.
3. Lines 306-309 are not relevant to this study.
4. The description in line 315 should have literature citations.
5. It is inappropriate to simply discuss supplements in line 399 and has nothing to do with the topic.
6. The ideas in the discussion part are very jumpy, and most of the citations are not related to the topic of this manuscript. It is recommended to use titles to classify and clarify the point of view.
7. In general, the title of the manuscript does not clearly express the true content of the manuscript.
Reviewer 4 Report
First of all, thank you for the invitation to the review.
Towards the authors I recommend:
Proper registration number of PROSPERO as they have entered an identifier of another topic ( stroke). This is an important error.
As for the search methodology, they comment on the keywords and operators used, but not on how they have introduced these words with their relevant operators and the results obtained in each methodology.
I encourage authors to check the PROSPERO register and the search methodology.